# The Complex Relationship Between Serum Uric Acid, Endothelial Function and Small Vessel Remodeling in Humans

**DOI:** 10.3390/jcm9072027

**Published:** 2020-06-28

**Authors:** Stefano Masi, Georgios Georgiopoulos, George Alexopoulos, Konstantinos Pateras, Javier Rosada, Gino Seravalle, Carolina De Ciuceis, Stefano Taddei, Claudio Borghi, Guido Grassi, Damiano Rizzoni, Agostino Virdis

**Affiliations:** 1Department of Clinical and Experimental Medicine, University of Pisa, 56124 Pisa, Italy; stefano.taddei@med.unipi.it (S.T.); agostino.virdis@med.unipi.it (A.V.); 2National Centre for Cardiovascular Prevention and Outcomes, Institute of Cardiovascular Science, University College London, London EC1A 4NP, UK; 3Department of Twin Research & Genetic Epidemiology, King’s College London, London WC2R 2LS, UK; 4School of Biomedical Engineering and Imaging Sciences, King’s College London, London WC2R 2LS, UK; georgiopoulosgeorgios@gmail.com; 5Department of Clinical Therapeutics, National and Kapodistrian University of Athens, 10679 Athens, Greece; 6Department of Statistics, Athens University of Economics and Business, 10434 Athens, Greece; gs.alexopoulos@gmail.com; 7Department of Biostatistics and Research Support, Julius Center for Health Sciences and Primary Care, University Medical Center Utrecht, 3584 CG Utrecht, The Netherlands; kostas.pateras@gmail.com; 8Fourth Unit of Internal Medicine, University Hospital of Pisa, 56124 Pisa, Italy; j.rosada@ao-pisa.toscana.it; 9Cardiology Unit, Fondazione Istituto Auxologico Italiano, Ospedale S. Luca, IRCCS Istituto Auxologico Italiano, 20149 Milan, Italy; g_seravalle@yahoo.com; 10Department of Clinical and Experimental Sciences, University of Brescia, 25121 Brescia, Italy; cdeciuceis@gmail.com (C.D.C.); Damiano.rizzoni@unibs.it (D.R.); 11Department of Medical and Surgical Science, Alma Mater Studiorum University of Bologna, 40126 Bologna, Italy; claudio.borghi@unibo.it; 12Clinica Medica, Deptartment of Health Science, University Milano-Bicocca, 20126 Milan, Italy; guido.grassi@unimib.it; 13Division of Medicine, Spedali Civili di Brescia, Montichiari Hospital, 25018 Brescia, Italy

**Keywords:** uric acid, microvascular, remodeling, endothelial function

## Abstract

Aims: The relationship between serum uric acid (SUA) and microvascular remodeling in humans remains largely unexplored. We assessed whether SUA provides additional information on the severity of microvascular remodeling than that obtained from the European Heart Score (HS), the patterns of microvascular remodeling associated with changes in SUA levels and the mediation by endothelial function and nitric oxide (NO) availability on this relationship. Methods: A total of 162 patients included in the microvascular dataset of the Italian Society of Hypertension with available information on SUA, media-to-lumen (M/L) ratio, media cross-sectional area (MCSA), endothelial function, NO availability and HS were included in the analysis. The top tertile of M/L ratio and MCSA were used to define severe microvascular remodeling. Results: A U-shaped association was observed between SUA and both M/L ratio and MCSA. Adjustment for HS did not affect these associations. SUA was able to reclassify a significant number of subjects without, and with, severe M/L ratio and MCSA remodeling over the HS alone. The microvascular remodeling associated with SUA levels presented a predominant hypertrophic pattern. SUA was inversely associated with endothelial function and NO availability. Structural equation modeling analysis controlling for the HS suggested that the association of SUA with M/L ratio and MCSA was mediated through changes in endothelial function and NO availability. Conclusions: The addition of SUA to the HS improves the identification of subjects with greater microvascular remodeling. The relationship between SUA and microvascular remodeling is mediated by endothelial function and NO availability.

## 1. Introduction

A number of large epidemiologic studies have demonstrated the predictive value of serum uric acid (SUA) on the risk of cardiovascular (CV) disease and mortality, independently from common CV risk factors [1,2,3]. Based on these findings, European Guidelines have included SUA among factors to be assessed for the CV risk stratification of patients with arterial hypertension [4].

Increased circulating levels of SUA are not only related to the risk of clinical events but also to more severe organ damage [5,6,7], suggesting that SUA might promote the whole evolution of CV remodeling, from its subclinical manifestations to its clinical complications. This is in keeping with the mechanisms by which SUA is thought to contribute to the evolution of CV disease that lies in increased levels of vascular wall oxidative stress and inflammation [8]. Given that microvascular remodeling is currently considered the first manifestation of CV damage and is associated with the risk of CV events [9,10,11,12,13], a potential impact of SUA on microvascular remodeling might explain its association with future clinical and subclinical manifestations of CV disease. However, while several experimental models of asymptomatic hyperuricemia have provided evidence of a potential impact of SUA on microvascular remodeling [14,15,16,17], the incremental prognostic role of SUA on top of established CV factors for the severity of microvascular disease in humans has not yet been established.

We recently documented that endothelial dysfunction and, more specifically, a reduced nitric oxide (NO) availability, are strongly and independently related to measures of microvascular remodeling [18]. This is likely related to the capacity of endothelial function and NO availability to inform on both the damaging signals of CV risk factors and the capacity of the subjects to respond to such injuries. Common final pathways by which CV risk factors might alter the endothelial function and NO availability is by promoting the production of reactive oxygen species (ROS) and inflammatory cytokines, alterations that are also detected in subjects with hyperuricemia [19]. Thus, the capacity of SUA to promote the evolution of subclinical microvascular damage might relate to its influence on endothelial function.

In this study, we used the large microvascular data set from the Italian Society of Hypertension including microvascular, SUA and cardiovascular risk factor data from more than 100 subjects to assess: (i) whether SUA can provide additional information on the severity of resistance artery remodeling than that obtained from traditional scores used to define the patient’s cardiovascular risk, (ii) the predominant pattern of small vessel remodeling associated with different SUA levels, and (iii) whether the relationship between SUA and measures of resistance artery remodeling is mediated by endothelial dysfunction and reduced NO availability.

## 2. Methods

### 2.1. Population

Analyses were performed using the population of the microvascular data set of the Italian Society of Hypertension who had available information on resistance artery structure and SUA levels. The characteristics of the data set have been previously described and reported in the Appendix A [20]. A diagram reporting the attrition of the population included in the Italian Society of Hypertension dataset based on the availability of the variables used to address each research question of this study is reported in the Appendix A. To address the first and second research question of this study, we used data collected from three research centers in Italy (Brescia, Pisa, and Milano), for a total of 162 subjects with available measures of media-to-lumen (M/L) ratio and SUA levels. Among these 52% of the participants had a diagnosis of arterial hypertension, which was made according to the clinical history of the patient and confirmed by a direct recording of blood pressure values >140/90 mmHg during the clinical visit or the use of antihypertensive medications. Furthermore, 70 out of the 162 participants with available M/L ratio and SUA also had measures of endothelial function, which were used to address the third research hypothesis. The microvascular data set of the Italian Society of Hypertension has already shown its ability to provide reliable information on the relationship of small resistance artery remodeling with levels of cardiovascular risk factor, endothelial function and NO availability in previous publications [18,20]. The studies that contributed to the data set were conformed to the principles outlined in the Declaration of Helsinki, they were approved by the local ethics committees in each center, and the informed written consent for subsequent analyses was obtained from each participant. 

### 2.2. Measurements

Information on the severity of small resistance artery remodeling, endothelial dysfunction and reduced NO availability were obtained using micromyography, as previously described [18,20]. Briefly, following isolation from biopsies of subcutaneous tissue collected from the anterior abdominal wall or gluteal regions, small resistance arteries were mounted on a pressure or wire myograph to assess the M/L ratio and media cross-sectional area. From the structural parameters the growth and remodeling indexes were also calculated to define the predominant pattern of remodeling. The primary measure used to assess vascular remodeling was the M/L ratio, as this has been shown to predict the individual’s cardiovascular outcome [12,13]. Details of the methods used to assess M/L ratio, growth index, and remodeling index are reported in the Appendix A. The distribution of participants with and without arterial hypertension was different among the group of subjects assessed with the wire (80% hypertensives) or the pressure (40% hypertensives) myography. In 70 subjects recruited in Pisa, the acquisition of the remodeling parameters was followed by the assessment of endothelial function and NO availability, as previously reported and described in the Appendix A. 

Serum Uric Acid was measured according to standardized protocols. Age, blood pressure values, hypertension status, body mass index (BMI), total cholesterol, high-density lipoprotein cholesterol, triglycerides, blood fasting glucose, and serum creatinine were collected and, for each patient, were used to calculate the heart score (HS) using the online tool provided by the European Guidelines on Cardiovascular Disease Prevention [21]. In addition, the estimated glomerular filtration rate (eGFR) was calculated using the Cockcroft–Gault formula.

### 2.3. Statistics

Data are expressed as mean ± standard deviation for normally distributed variables or median (25–75%) for not normally distributed variables. The maximal percentage increments or decrements of lumen diameter from baseline obtained from the dose-response curves to acetylcholine was used as a measure of endothelial function for statistical analyses. Initially, linear regression models were used to test the association between SUA and parameters of microvascular remodeling (M/L ratio and media cross-sectional area (MCSA)), endothelial function and NO availability. Indices of microvascular remodeling were used as dependent variables in the regression analysis after natural log transformation to decrease skewness and extreme observations influence.

To capture the nonlinear relationship between SUA levels and M/L ratio, we employed fractional polynomial analysis and considered various degrees of polynomial transformations for SUA. The best fitting second degree polynomial was selected among power transformations from the set, where 0 denotes the log transformation. The best fitting second-degree fractional polynomial was compared against simpler transformations of SUA model using a deviance difference test. For both M/L and MCSA, a second-degree fractional polynomial with a linear and a quadratic term of SUA presented the optimal fit to the observed data. The same approach of fractional polynomial regression was employed to explore the association between levels of SUA and parameters defining the prevalent remodeling patterns (i.e., growth and remodeling index); a second-degree fractional polynomial with a linear and a quadratic term of SUA was selected for this analysis as previously described after standardization of the growth index and remodeling index. Given that an impaired renal function could influence the relationship between SUA and microvascular remodeling, these analyses were repeated excluding subjects with eGFR < 60 mL/min.

The additive value of SUA over the HS for classifying the severity of resistance artery remodeling in terms of M/L ratio and MCSA was estimated by the continuous net reclassification improvement [22]. A dichotomous outcome (severe remodeling defined as being classified in the highest tertile of M/L or MCSA versus lower tertiles) was used in the net reclassification improvement analysis.

Subsequently, we assessed the attenuation of the relationships between SUA and measures of microvascular remodeling after adjustment for the potential mediators (endothelial function or NO availability). Such an attenuation was considered suggestive of a mediating effect of endothelial measures on the association between SUA and M/L ratio or MCSA. To confirm the possible mediating effect of endothelial function or NO availability on the relationships between SUA and microvascular remodeling and its significance we estimated the average causal mediation effect (ACME) by running 1000 bootstrap resamples with replacement and used structural equation modelling (SEM). Estimated ACME, average direct effect (ADE) and total effect are provided as median and 95% CI after bootstrapping. All SEM model parameters were estimated by the full information maximum likelihood estimation (FIML) method, with respect to the missing data. We employed maximum likelihood estimation with “robust” (Huber–White) standard errors, and a robust test statistic (Satorra–Bentler chi-square test) for model evaluation to tackle deviation from normality in variables inserted in the SEM models. To evaluate the overall goodness of fit for the structural equation models, we used the model chi-square statistic test, and the root mean square error of approximation with values lesser or equal to 0.1 indicating good fit. Structural equation models were visualized with path diagrams. Statistical analyses were performed using Stata ver. 13.1 (StataCorp, College Station, TX, USA); the modules “fp” and “mfp” were used for univariable and multivariable fractional polynomial analysis, respectively. The lavaan package was used in *R* ver. 3.5.3 to implement the SEM models. All statistical tests were two-tailed; we set the level of statistical significance at *p* < 0.05.

## 3. Results

The characteristics of the 162 study participants with available SUA levels are reported in Table 1, stratified by tertiles of SUA. Prevalence of male gender, levels of triglycerides, ML, MCSA, and HS increased across ascending tertiles of SUA, while NO and acetylcholine (Ach) decreased (Table 1).

### 3.1. Relationship Between Uric Acid Levels and Microvascular Remodeling with and without Controlling for Established Cardiovascular Risk Factors

We observed a U-shape association of SUA levels with M/L ratio: low (β = 0.139 unit change in log M/L ratio; 95% confidence interval (CI) −0.001, 0.280; *p* = 0.051) and high tertile (β = 0.185 unit change in log M/L ratio; 95% CI 0.041, 0.328; *p* = 0.012) of SUA, were related to increased M/L ratio as compared to the middle (reference) tertile. Fractional polynomial analysis confirmed the non-linear association of continuous SUA with M/L ratio (β = −0.354; 95% CI −0.557, −0.152; *p* < 0.001 for the linear term and β = 0.031; 95% CI 0.014, 0.047; *p* < 0.001 for the non-linear, squared term) (Figure 1A). Similarly, SUA levels showed that a non-linear association with MCSA: low (β = 0.142 unit change in log M/L ratio; 95% CI −0.033, 0.317; *p* = 0.112) and high tertile (β = 0.246 unit change in log M/L ratio; 95% CI 0.067, 0.426; *p* = 0.007) of SUA, were related to increased MCSA as compared to the middle (reference) tertile. Fractional polynomial analysis confirmed the non-linear association of continuous SUA with MCSA (β = −1.59; 95% CI −2.22, −0.952; *p* < 0.001 for the linear term and β = 0.581; 95% CI 0.351, 0.81; *p* < 0.001 for the non-linear, squared term) (Table 2) (Figure 1B). 

The linear and non-linear associations between SUA and parameters of microvascular remodeling were significant in both sexes (Appendix A), in patients with and without hypertension (Appendix A) and exclusion of patients with eGFR < 60 mL/min did not affect the results (Appendix A). Furthermore, adjustment for the HS did not attenuate the significant relationship of SUA with M/L ratio and MCSA (*p* < 0.001 for both). 

SUA levels showed a hyperbolic association with growth index (standardized β = 25.6, *p* = 0.01 for the linear term and standardized β = −24.6, *p* = 0.014 for the non-linear, squared term) (Appendix A). An inverse parabolic association was found between SUA levels and the remodeling index (standardized β = 1.08, *p* = 0.021 for the linear term and standardized β = −0.078, *p* = 0.041 for the non-linear, squared term) (Appendix A). Given that we found a significant difference in the growth index between wire and pressure-myography, the analysis on the prevalent pattern of remodeling associated with SUA was repeated stratifying the population based on the method used for the microvascular assessment. As reported in the Appendix A, also the stratified analysis documented a prevalent pattern of hypertrophic remodeling associated with SUA, growth given that the growth index remained highly significantly associated with SUA in both models, while remodeling index lost its significance.

By reclassification analysis, SUA correctly reclassified 7.14% and 45.16% of subjects without and with severe remodeling using M/L ratio over HS (NRI = 0.523, SE = 0.218, *p* = 0.016). Substantial reclassification improvement was obtained with MCSA as well. Indeed, the addition of SUA to the HS led to a correct reclassification of 25% and 47.27% of subjects without and with severe MCSA remodeling, respectively; thus, accounting for a total NRI of 0.727, SE = 0.23, *p* = 0.002.

### 3.2. Relationship Between Uric Acid Levels and Microvascular Remodeling: the Effect of Endothelial Function

SUA was also associated with endothelial function and NO availability, so that subjects with higher SUA had a reduced endothelial-dependent vasodilation (β = −4.86 unit change in max vasodilation to Ach per mg/dl increase in SUA; 95% CI −7.91, −1.80; *p* = 0.003). An even stronger relationship was identified, as expected, between SUA and levels of NO availability (β = −5.72 unit change in NO availability per mg/dl increase in SUA; 95% CI −8.44, −2.99; *p* < 0.001). These associations remained highly significant when the analyses were restricted to subjects with a normal eGFR (Appendix A). The addition of endothelial function as independent variable led to a substantial attenuation of the association between SUA and both parameters of microvascular remodeling (Table 2). Similar results were obtained when NO availability was used instead of endothelial function (Table 2).

### 3.3. Structural Equation Modelling

By SEM analysis and after controlling for the clustering of established cardiovascular risk factors (Heart Score), SUA was positively associated indirectly with M/L ratio (ACME = 0.28 per mg/dl increase in SUA; 95% CI 0.14, 0.43; *p* < 0.001) and MCSA (ACME = 0.35 per mg/dl increase in SUA; 95% CI 0.22, 0.48; *p* < 0.001), through changes in endothelial function. A direct effect was not established (for M/L ratio: ADE = 0.08; 95% CI −0.05, 0.21; *p* = 0.22 and for MCSA: ADE = 0.01; 95% CI −0.16, 0.17; *p* = 0.94) (Table 3, Figure 2A,B). Both SEM models presented satisfactory fit to data (RMSE ≤ 0.1 and chi-squared test *p* > 0.05).

Accordingly, NO availability conferred significant mediation effect on the association of SUA with M/L ratio (ACME = 0.35 per mg/dl increase in SUA; 95% CI 0.22, 0.48; *p* < 0.001) and MCSA (ACME = 0.34 per mg/dl increase in SUA; 95% CI 0.20, 0.48; *p* < 0.001) (Table 3, Figure 2C,D). SEM analysis did not demonstrate significant direct effect of SUA on indices of microvascular remodeling (for M/L ratio: ADE = 0.02; 95% CI −0.13, 0.17; *p* = 0.80 and for MCSA: ADE = −0.05; 95% CI −0.24, 0.14; *p* = 0.60) (Figure 2C,D). SEM fitting of the mediating effect of NO availability on M/L ratio was satisfactory (*p* = 0.45) but suboptimal for MCSA (*p* = 0.023, RMSE = 0.19).

## 4. Discussion

This is the first study to document a clear U-shaped association between SUA and parameters of microvascular remodeling in humans. Low and high levels of SUA were associated with hypertrophic remodeling of the small resistance vessels, characterized by higher M/L ratio, MCSA, and growth index. The observed association was independent of other established CV risk factors, and the addition of SUA to the HS was able to reclassify a significant number of patients with severe and non-severe microvascular remodeling. Importantly, the effects of SUA on both M/L ratio and MCSA were mediated by endothelial function and NO availability. In summary, our results suggest that the measure of SUA might provide important additive information on the severity of microvascular remodeling than the HS alone. They also indicate that the assessment of endothelial function might represent an early and accurate marker to monitor the potential vascular benefits obtained with urate-lowering treatment. 

An impact of SUA on microvascular remodeling has been described in animal models and cell culture experiments of hyperuricemia, where SUA can induce smooth muscle cell proliferation leading to increased MCSA and M/L ratio [14,16]. We now provide evidence that SUA levels is strongly associated with measures of small vessel remodeling also in humans and that the addition of SUA to the estimation of total CV risk with the use of the HS improves the capacity to identify patients with more or less severe small vessel remodeling. The U-shaped relationship between SUA and both M/L ratio and MCSA observed in our study suggests that also very low levels of SUA might promote the evolution of microvascular disease. This is in keeping with the evidence that at physiological concentrations, SUA is a potent antioxidant in vitro [23,24] and might have a protective effect on the vasculature. Therefore, low concentrations of SUA, particularly in extracellular fluids, could result in reduced antioxidant capacities, whilst high concentrations would promote its intracellular translocation, activating proliferative and pro-oxidant inflammatory pathways. The capacity of SUA to influence endothelial function due to its antioxidant effects is supported by in vivo studies. Indeed, uric acid administration improves endothelial function in the forearm vascular bed of patients with type 1 diabetes and in smokers. The extend of such improvements is similar to what can be obtained with the infusion of ascorbic acid, a potent antioxidant [25]. Further support on the capacity of uric acid to positively influence endothelial function is derived from patients affected by renal hypouricemia, a genetic disorder characterized by impaired reabsorption of uric acid and consequently low levels of SUA. Sugihara et al. reported that in patients with renal hypouricemia and SUA levels < 2.5 mg/dl endothelial function assessed by the flow-mediated dilation is correlated with SUA levels, and that patients with SUA < 0.8 mg/dL had a more pronounced impairment of the flow-mediated dilation than those with SUA between 0.8–2.5 mg/dL [26]. Compared to these reports, we now show that the negative effect of low levels of SUA on endothelial function might translate in higher vascular remodeling, potentially leading to an increased risk of cardiovascular disease. Indeed, several observational studies have documented a U- or J-shaped relationship between SUA and risk of CV events or mortality [27,28,29,30,31,32,33].

Although in vitro experiments have suggested that SUA might directly stimulate proliferation of smooth muscle cells [16,34], our data suggests that on intact human small resistance arteries endothelial function and, more specifically, NO availability might play a key role in mediating the remodeling signal triggered by changes in the SUA levels. In cultured human endothelial cells, high concentrations of uric acid promote its translocation within the intracellular space, reducing NO availability by increasing ROS production [19] and through direct and irreversible reaction with the NO resulting in the formation of 6-aminouracil [35]. This is accompanied by the activation of vascular remodeling systems, such as the renin-angiotensin axis, which can further promote microvascular remodeling [19]. It is therefore possible that in intact vessels the endothelium represents the primary signal transducer of a high SUA concentration in the bloodstream, representing the intermediate mediator of its signals. Although our data support full mediation by NO availability and endothelial function on the relationship between SUA and small resistance artery remodeling, they do not exclude that in diseased conditions characterized by loss of endothelial integrity (i.e., in the context of atherosclerotic plaques) elevated concentrations of SUA might contribute to vascular remodeling by direct effects on other components of the vascular wall. Indeed, uric acid crystals have been detected within the vascular wall of atherosclerotic vessels [36]. Another potential explanation of our findings might lie in the capacity of SUA to mark the hyperactivity of the xanthine oxidase, the enzyme catalyzing the final reactions of the uric acid production, whom activity is also coupled with the production of ROS. Following this hypothesis, subjects with elevated SUA levels might represent those with an overactivity of the xanthine oxidase, leading also to increased production of ROS, endothelial dysfunction and, through this pathway, greater microvascular remodeling. Beyond the potential biological mechanisms underlying the relationship between SUA, microvascular remodeling, and endothelial function, our results support the use of endothelial function as an early indicator of the potential cardiovascular benefits achievable with urate-lowering treatments.

The low and high levels of SUA that show the strongest association with the M/L ratio are related to a predominant pattern of hypertrophic remodeling, with a prevalent association with the growth rather than the remodeling index. The association of high SUA levels with hypertrophic remodeling is in keeping with previous animal and cultured experiments, showing that SUA can promote smooth muscle cell proliferation leading to hypertrophy of the media microvascular layer [14,16]. Conversely, the strong association evidenced in our study between low SUA levels and hypertrophic remodeling has not been described previously in humans. We can only speculate on the biological mechanisms accounting for this association. For example, the lack of adequate antioxidant defenses in subjects with low SUA levels might result in a deficit of NO availability which, in turn, has been associated with a prevalent pattern of hypertrophic rather than eutrophic remodeling [18]. 

The microvascular dataset of the Italian Society of Hypertension has some unique features for the study of the relationship between parameters of vascular remodeling, endothelial function, and CV risk factors. The large sample size for a microvascular study and availability of data on a wide range of established risk factors and potential confounders enabled the independent role of many parameters to be assessed. The micromyography is currently considered the gold standard technique to assess resistance artery structural changes, and the database of the Italian Society of Hypertension has already proven its reliability for analyses assessing the complex relationships between CV disease risk factors and parameters of microvascular remodeling [18,20]. The M/L ratio, used as primary endpoint of our analysis, is a validated measure of microvascular remodeling and has been shown to predict cardiovascular events in populations at high and low cardiovascular risk [12,13]. Previous publications have shown a substantial overlap of the M/L ratio obtained with the pressure and wire micromyography [37]. In addition, its measure is far less influenced by possible sampling bias, an issue that could affect other parameters of microvascular structure, such as the lumen diameter and the media. Finally, the availability of information on endothelial function and NO availability in a large subsample of participants enabled a proper mediation analysis to be conducted, largely enhancing our understanding of the mechanisms underpinning the relationship between SUA, endothelial dysfunction, and vascular remodeling.

Our work also has important limitations. The observational and cross-sectional study design do not provide the opportunity to assign/infer causality from our data reliably. However, it should be emphasized that longitudinal studies using the micromyography technique are limited by the availability of repeated collection of biopsy samples due to ethical constraints. Nevertheless, despite having cross-sectional data, we were able to conduct a formal mediation analysis to identify the direction of the associations between SUA, remodeling parameters, endothelial function, and NO availability. We did not have information on measures of vascular wall oxidative stress, making any explanation on the mechanisms potentially accounting for the endothelial dysfunction in participants with hyperuricemia speculative. The report lacks hard cardiovascular endpoints. However, the scope of our study was not to confirm the relationship between SUA and CV outcomes, rather it was aimed at identifying potential mechanisms underpinning this associations and the measure of M/L ratio has been previously proven to be a potent predictor of CV endpoints, independently from the levels of common CV risk factors. Given that the microvascular dataset of the Italian Society of Hypertension was established by merging data from studies conducted in order to answer different research questions, some important information potentially influencing the relationship between SUA and microvascular remodeling were missing. Among these, we did not have information on the prior diagnosis of gout, making it impossible to assess whether the impact of hyperuricemia on parameters of microvascular remodeling/function differs among subjects with or without grout. Moreover, we did not have information on treatments; thus, we could not assess the effect of urate lowering drugs, or antihypertensive drugs, on the relationship between SUA and microvascular remodeling parameters. 

In conclusion, this study suggests that the addition of SUA to the assessment of common CV risk factors might significantly improve the capacity to identify people at increased risk of microvascular remodeling in the general population. Given that small vessel disease is considered the earliest organ damage related to CV risk factor exposure, this relationship might explain the associations between SUA and CV events described in several large observational studies. In humans without atherosclerosis, the influence of SUA on the vascular phenotype is likely mediated by its impact on endothelial function and NO availability.

## Figures and Tables

**Figure 1 jcm-09-02027-f001:**
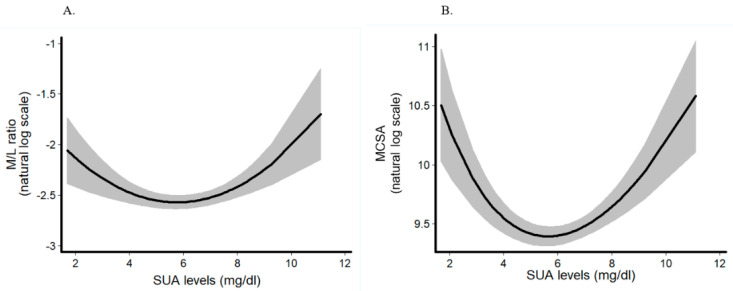
Fractional polynomial regression analysis for the association between continuous levels of serum uric acid (SUA) and (**A**) media-to-lumen ratio (M/L ratio) (*n* = 162 subjects) and (**B**) media cross-sectional area (MCSA) (*n* = 157 subjects). Serum uric acid is modelled as a second-degree fractional polynomial. A second-degree fractional polynomial with a linear and a quadratic term of SUA presented the optimal model fit to describe the relationship between SUA levels and remodeling parameters.

**Figure 2 jcm-09-02027-f002:**
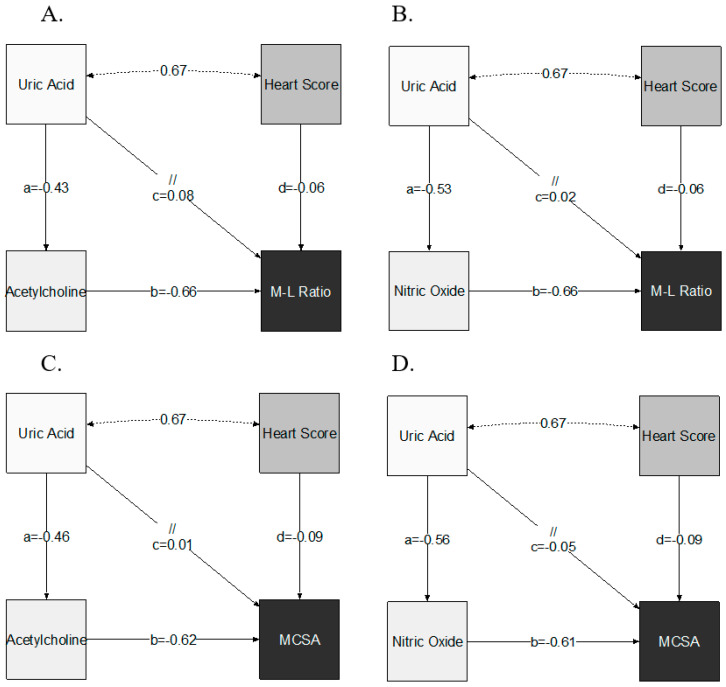
Path diagrams for structural equation models of the mediating effect of endothelial function ((**A**,**C**): Acetylcholine; (**B**,**D**): NO availability) on the association between uric acid levels and indices of microvascular remodeling (A and B: M/L ratio; C and D: MCSA). Endothelial function was available in 70 subjects. In each path diagram, independent variables have unidirectional arrows pointing to the dependent variable. The parameter coefficient that has been estimated by the SEM models is placed adjacent to corresponding arrows. Coefficients a and b indicate the indirect effect (a*b) whilst coefficient c quantifies the direct effect. Double headed arrows represent the covariance estimation between uric acid levels and the Heart Score.

**Table 1 jcm-09-02027-t001:** Characteristics of the population stratified by tertiles of serum uric acid (SUA).

Variable	All (SD)	1st Tertile of SUA	2nd Tertile of SUA	3rd Tertile of SUA	*p*-Value
**N**	**162**	**57**	**53**	**52**	
Age (years)	46 (14)	45 (14)	45 (14)	51 (13)	0.067
Gender (male, %)	88 (54)	24 (42)	26 (49)	38 (73)	0.003
Smoking (never, %)	34 (26)	10 (20)	13 (35)	11 (25)	0.401
Serum Uric Acid (mg/dl)	6.1 (1.82)	4.7 (1.1)	6.1 (0.5)	7.3 (0.95)	<0.001
Systolic blood pressure (mmHg)	142 (18)	139 (17)	144 (21)	142 (16)	0.456
Diastolic blood pressure (mmHg)	88 (11)	86 (11)	89 (12)	89 (11)	0.228
HDL-Cholesterol (mg/dl)	47 (10)	49 (11)	48 (10)	44 (9)	0.06
LDL-Cholesterol (mg/dl)	122 (25)	113 (24)	124 (26)	126 (25)	0.078
Glucose (mg/dl)	97 (24)	97 (26)	97 (26)	97 (19)	0.989
Creatinine (mg/dl) *	0.88 (0.34)	0.83 (0.23)	0.96 (0.32)	0.87 (0.35)	0.004
eGFR (ml/min) *	105 (63.7)	120 (72)	94.6 (28.5)	105 (118)	0.011
Diagnosis of Dyslipidemia (%)	58 (53)	17 (41)	15 (56)	26 (63)	0.132
Diagnosis of Hypertension (%)	84 (52)	29 (51)	27 (51)	28 (54)	0.941
Diagnosis of diabetes mellitus, *n* (%)	19 (12)	11 (20)	5 (10)	3 (6)	0.072
M/L ratio *	0.079 (0.042)	0.082 (0.042)	0.063 (0.036)	0.084 (0.04)	0.008
MCSA (µm^2^) *	12,000 (11,364)	12,000 (12,000)	10,000 (7200)	13,000 (12,000)	0.019
Lumen (µm)	220 (52)	235 (60)	220 (46)	215 (37)	0.021
Lumen (pressurized myography) (µm)	218 (39)	221 (42.6)	215.5 (27)	217.1 (48)	
Lumen (wire myography) (µm)	250 (91)	266.41 (111.71)	252.05 (76.49)	174.8 (37.88)	
Media(µm)	16.7 (10.9)	17.5 (12.5)	13.9 (9.1)	19.5 (11,6)	0.007
Media (pressurized myography) (µm)	14.5 (9.45)	13.4 (7,5)	13.2 (2.2)	18.72 (11.24)	
Media (wire myography) (µm)	21.5 (8.24)	28.89 (8.0)	23.15 (6.79)	19.93 (11.28)	
Remodeling index *	999 (2409)	987 (2100)	1100 (1000)	929 (2800)	0.834
Growth Index **	30.9 (12.13)	30.9 (12.1)	30.9 (12.1)	36 (26.2)	0.521
Endothelial function (%) *	74.5 (15)	76 (19)	75.3 (18.5)	72 (6.8)	0.035
NO Availability **	15 (13.3)	17 (11.5)	15.4 (14.8)	13 (6)	0.026
Heart Score **	0 (1)	0 (1)	0 (1)	1 (2)	0.0025

* Median and interquartile range. Abbreviations: HDL, high-density lipoprotein; LDL, low-density lipoprotein; eGFR, estimated glomerular filtration rate; and endothelial function, % of maximal vasodilation compared to the baseline vascular diameter obtained with acetylcholine infusion. ** Non-normally distributed variables.

**Table 2 jcm-09-02027-t002:** Change of the relationship between SUA and parameters of vascular remodeling following adjustment for endothelial function and nitric oxide (NO) availability.

	Unadjusted		Adjusted for Endothelial Function		Adjusted for NO Availability	
	Coefficient (95% CI)	*p*-Value	Coefficient (95% CI)	*p*-Value	Coefficient (95% CI)	*p*-Value
**Dependent variable: M/L ratio**						
SUA	−0.354 (−0.557, −0.152)	<0.001	−0.017 (−0.034, 0.067)	0.511	0.012 (−0.04, 0.063)	0.647
**Dependent variable: MCSA**						
SUA	−1.59 (−2.22, −0.952)	<0.001	−0.015 (−0.097, 0.067)	0.716	−0.027 (−0.113, 0.06)	0.541

Abbreviations: M/L ratio = media-to-lumen ratio; MCSA = media cross-sectional area; SUA = serum uric acid. Fractional polynomials were used to model the association of continuous uric acid with M/L ratio and MCSA. Coefficients correspond to first degree polynomials or the linear term of second-degree polynomials. M/L ratio and MCSA were used in the natural log transformed scale. Adjusted models additionally controlled for Heart Score.

**Table 3 jcm-09-02027-t003:** Estimation of the average direct effect (ADE) of serum uric acid levels and average causal mediation effect (ACME) through endothelial function and nitric oxide (NO) availability on parameters of microvascular remodeling. The average causal mediation effect (ACME) is the indirect effect of SUA on the outcomes through the mediator variables after controlling for Heart Score effect. The average direct effect (ADE) represents the direct effect exclusively explained by SUA on the remodeling indices.

Dependent Variable	Mediator	Metric	Estimate	95% CI Lower	95% CI Upper	*p*-Value
M/L ratio	Endothelial Function (Ach)	ACME	0.28	0.14	0.43	<0.01
		ADE	0.08	−0.05	0.21	0.22
		Total Effect	0.37	0.21	0.52	<0.01
M/L ratio	NO Availability	ACME	0.35	0.22	0.48	<0.01
		ADE	0.02	−0.13	0.17	0.80
		Total Effect	0.37	0.21	0.52	<0.01
MCSA	Endothelial Function (Ach)	ACME	0.28	0.14	0.43	<0.01
		ADE	0.01	−0.16	0.17	0.94
		Total Effect	0.29	0.13	0.45	<0.01
MCSA	NO Availability	ACME	0.34	0.20	0.48	<0.01
		ADE	−0.05	−0.24	0.14	0.60
		Total Effect	0.29	0.14	0.45	<0.01
Unstandardized effects were computed after bootstrapping with 1000 replicates						

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
