# Peer review of "The Complex Relationship Between Serum Uric Acid, Endothelial Function and Small Vessel Remodeling in Humans"

_jcm, 2020, doi:10.3390/jcm9072027_

Round 1

Reviewer 1 Report

This study is interesting and shows complex association between serum uric acid concentrations with  microvascular beds dysfunction characterized by NO and endothelial function. 

This paper is well written and overall demonstrates novel findings. 

Major points:

1-Since high levels of uric acid are associated with gout, it is important to determine individuals with gout in the cohort and reanalyze data stratified by this condition.

2-Also, the profiles of SUA in females in not well defined. Analyzing SUA in women would provide further insights into its profiles across gender.

Minor point: too many abbreviations.

Author Response

We thank the reviewer for the positive and constructive comments that substantially improved the quality of our manuscript. We responded to all reviewer comments (please see the attachment), performing additional analyses, providing clarifications and, where needed, adding references to support our statements. We highlighted changes made to the main text and supplementary material based on the reviewer comments.

Reviewer 2 Report

Masi et al. in manuscript “The complex relationship between serum uric acid, endothelial function and small vessel remodeling in humans” studied the relationship between serum uric acid (SUA) and microvascular remodeling in humans. The author found that a U-shaped associated between these two factors. The author conclude that both low and high SUA levels promote ROS production leading to EC dysfunction and deficit of NO availability. These responses further increases media-to-lumen ration and media cross-sectional area. They summarize that measuring SUA level could be used to predict microvascular remodeling. This study was well performed and found a potential candidate, the SUA, for evaluation of microvascular remodeling.

The concern is how SUA could function as anti-oxidative stress agent and pro-oxidative agent with different concentration. Especially, the rationale of linking low SUA level-induced microvascular remodeling to loss of SUA’s anti-oxidative stress activity is not convincing, because there are other anti-oxidative stress systems (e.g., GSH, GPx, SOD). If no data can support this accociation, more discussion is needed to address the significance of the U-shaped association between SUA and vascular function.

Author Response

We thank the reviewer for its positive comments to our work. Please see the attachment for a detailed response to the reviewer comments/questions.

Reviewer 3 Report

The paper by Masi et al describes the relation between serum uric acid (SUA) and small vessel structure and endothelial function. This is an interesting manuscript with new and potential important observations. I have some suggestions and questions. I also have some concerns that need to be addressed.

  1. In the introduction you use the phrase “NO availability”; what is exactly meant by this term?
  2. I am aware that patient characteristics may have been presented before, but it is still important to have information about the participants. Concerning the 162 participants:
    1. How many were healthy controls?, how many had essential hypertension?, Were there any diabetics or patients with secondary hypertension?
    2. Medication: please include information about serum urate lowering medications (allopurinol and others) as well as antihypertensive medication.
    3. You give heart score (HS) in table 1, but this is low and not really informative. I suggest you include data on all elements of HS inclusive creatinine (or even better a calculated eGFR). Please mention how HS is calculated.
    4. You excluded stage 4-5 CKD; why not stage 2-3? CKD is strongly associated to increased SUA.
    5. Patients with previous CV events were also excluded, why?
    6. To get a better impression of the population (n=162) selected to participate in the present analysis I suggest you include a diagram giving the original number of patients in the SIIA database and how many were excluded due to various circumstances.
    7. The actual SUA concentrations should appear in the columns in table 1.
    8. Please also include vessel morphology characteristics (media, lumen, M/L, RI and GI) in table 1.
  3. Vessels were investigated by either wire or pressure myography. In contrast to a wire-mounted vessel segment, a pressurized vessel segment is elongated.
    1. Please mention media and lumen dimensions for wire- and pressurized vessels.
    2. Did they have similar passive characteristics; i.e. were the tension-diameter curves similar.
    3. Were there differences in RI and GI between pressure- and wire-mounted vessel segments (although one should not think so it should be stated).
    4. I assume the vessels from the “anterior abdominal wall” came from participants having some kind of abdominal surgery ? please include a little more information about this. Could their condition affect SUA somehow – did participants with gluteal biopsies have SUA in the same range as participants with abdominal biopsies ?
  4. The analyses are very complicated and a bit hard to comprehend. As far I understand adjustment for HS does not change the relation between M/L and SUA.
    1. However, as mentioned above HS is low (0 or 1). Would you then expect adjustment for HS to affect the relation?. What happens if you adjust for actual BP or eGFR?
    2. What happens if you include urate-lowering and antihypertensive medication in the analysis?
    3. Are the association influenced by the research methodology (wire or pressure myograph, or whether the vessels originate from the abdominal or gluteal region).
    4. In Figure 1 you illustrate M/L and MCSA as function of SUA. Please also include the association between SUA and lumen diameter.
  5. As mentioned in the discussion the mechanistic insight from this analysis is limited, but relevant topis have been discussed.
    1. Is it possible to discuss factors responsible for the interindividual variation in SUA. Can factors related to formation or metabolism (excretion) of urate be related to small artery remodelling ? Does fasting pay a role for SUA in this study?
    2. I wonder if there could be any methodological pitfalls that need to be discussed. I would assume lumen diameter to be independent of SUA; when the lab technician dissects vessels out of the fat biopsy these should be of similar diameters from all patients (as you look for vessels of a certain size) and therefore not dependent on variation in certain metabolic parameters like SUA. If this is not the case reasons for this must be discussed.

Author Response

(The authors gave the same response as above.)

Round 2

Reviewer 1 Report

Satisfied with revision. 

Author Response

We thank the reviewer for the positive comment to our revision that does not require further response.

Reviewer 3 Report

The extensive revision has improved the paper considerably and most issue are now solved.

However, I still believe the description of the study population could be improved. Table 1 states that 52% have a diagnosis of hypertension.

1) How is hypertension defined in this population ?   is it only based on information from the participant or is it based on blood pressure measurements. With no information about medication a normal BP does not exclude that the patient has hypertension.

2) How was BP measured in this study ?  are the values in table 2 office readings ?  and what were the measurement circumstances ?

3) Was the distribution between hypertensive and normotensive participants equal in the two centers (using wire- and pressure myography respectively) ?

This information could be included on page 3 in description of the population.

Furthermore:

4) If 51% had hypertension, I assume 49% were considered normotensive. Thus, the dataset is a mix of hypertensive and normotensive patients. Is the relation between SUA and microvascular structure (M/L, MCSA etc.) similar in the hyper- and normotensive subgroups ?  Should the analysis be adjusted for the presence or absence of hypertension ?

Author Response

We thank the reviewer for the additional comments. We have addressed all comments providing further explanations and performing the analyses requested by the reviewer. We hope that, following these clarifications and analyses, our manuscript could be considered for publication.
